# The Clinical Effectiveness and Tolerability of Oseltamivir in Unvaccinated Pediatric Influenza Patients during Two Influenza Seasons after the COVID-19 Pandemic: The Impact of Comorbidities on Hospitalization for Influenza in Children

**DOI:** 10.3390/v16101576

**Published:** 2024-10-07

**Authors:** Gheorghiță Jugulete, Mihaela Cristina Olariu, Raluca Stanescu, Monica Luminita Luminos, Daniela Pacurar, Carmen Pavelescu, Mădălina-Maria Merișescu

**Affiliations:** 1Faculty of Medicine, University of Medicine and Pharmacy, “Carol Davila”, No. 37, Dionisie Lupu Street, 2nd District, 020021 Bucharest, Romania; gheorghita.jugulete@umfcd.ro (G.J.); mihaela.olariu@umfcd.ro (M.C.O.); luminita.luminos@umfcd.ro (M.L.L.); daniela.pacurar@umfcd.ro (D.P.); carmen.pavelescu@rez.umfcd.ro (C.P.); madalina.merisescu@umfcd.ro (M.-M.M.); 2“Matei Balş” National Institute for Infectious Diseases, No. 1, Calistrat Grozovici Street, 2nd District, 021105 Bucharest, Romania; 3Department of Pediatrics, “Grigore Alexandrescu” Emergency Clinical Hospital for Children, No. 30-32, Iancu de Hunedoara Blvd., 011743 Bucharest, Romania

**Keywords:** oseltamivir, tolerability, influenza, children, clinical effectiveness, comorbidities

## Abstract

Antiviral therapy such as oseltamivir has been recommended for hospitalized children with suspected and confirmed influenza for almost 20 years. The therapy is officially authorized for newborns two weeks of age or older, however, questions about its safety and effectiveness still surround it. Our goals were to assess the epidemiological features of two consecutive seasonal influenza cases in children following the COVID-19 pandemic; to observe the clinical effectiveness and tolerability of oseltamivir in hospitalized children who were not vaccinated against influenza and had different influenza subtypes, including A(H1N1), A(H3N2), and B; and to identify specific comorbidities associated with influenza in children. We performed an observational study on 1300 children, enrolled between 1 October 2022 and 30 May 2023 and between 1 October 2023 and 4 May 2024, to the IX Pediatric Infectious Diseases Clinical Section of the National Institute of Infectious Diseases “Prof. Dr. Matei Balș”. During the 2022–2023 influenza season, 791 pediatric patients tested positive for influenza and received oseltamivir. Of these, 89% (704/791) had influenza A, with 86.4% having subtype A(H1N1) and 13.6% of cases having A(H3N2), and for influenza B, 11% (87/791) of the pediatric patients. Of the total group, 59% were male, and the median age was 2.4 years (1.02–9.28). For the 2023–2024 influenza season, 509 pediatric patients tested positive for influenza, with 56.9% being of the male gender and who were treated with oseltamivir. Of these patients, 81.6% had influenza A and 18.4% had influenza B. Treatment with neuraminidase inhibitors, specifically oseltamivir, 2 mg/kg/dose administered twice daily for 5 days, was well tolerated by the children, and we recorded no deaths. The duration of hospitalization for patients with a fever after the oseltamivir administration was significantly longer for patients with A(H1N1) infection than A(H3N2), during both seasons. We identified more complications in the 2022–2023 season and a decreasing number of influenza B for the 2023–2024 season. Among children with comorbidities, the most common were asthma, gastrointestinal diseases, and metabolic and endocrine diseases. In terms of effectiveness, oseltamivir significantly reduced the intensity of influenza symptoms, thus reducing the number of days of hospitalization (*p* = 0.001) as well as post-infection complications (*p* = 0.005) in both groups. In this study, we evaluated the clinical effectiveness of oseltamivir therapy for all influenza types/subtypes in children, and the length of hospitalization. We identified comorbidities associated with the prolonged duration of hospitalization. Influenza vaccination should be the main tool in the prevention of influenza and its complications in children, especially those with comorbidities.

## 1. Introduction

Every year, seasonal influenza epidemics and pandemics are brought on by influenza A(H1N1)pdm09, influenza A(H3N2), and influenza B, which severely threaten the health of young people and have been connected to a considerable number of hospitalizations in children and adolescents, and death in children under the age of five [1,2,3].

Seasonal influenza epidemics can result in 3–5 million severe cases globally and annually, with 650,000 influenza-related fatalities linked to respiratory diseases [3,4]. The majority of these either show no symptoms at all or was associated with fever, cough, nasal congestion, myalgia, headache, and gastrointestinal symptoms [3,4,5].

Comorbidities affect the severity of respiratory infections, as the COVID-19 pandemic has shown [3,4,6]. As SARS-CoV-2 and influenza are both respiratory illnesses that produce pandemics, vaccination efforts against influenza are a crucial part of prevention [6]. It is also critical to research how comorbidities affect the severity of influenza [7].

The population’s vulnerability to the influenza virus influences how each influenza season evolves in terms of severity, vaccination uptake, and duration [5,6,7,8]. Although influenza vaccinations have become more widely available and of higher quality, and the benefits of the disease in terms of lessening its effects on health and the economy are widely acknowledged, the majority of nations do not utilize enough vaccines to effectively control influenza [7,8,9]. Regarding the region, the cold season of 2022/2023 was dominated by viral infections in the pediatric population in Bucharest, Romania [2,4,7,8,9,10,11]. The evolution of acute respiratory infections, influenza, and severe acute respiratory infections, as presented by the National Institute of Public Health from the Ministry of Health, Romania, reports that at the end of February 2023, 81 deaths were due to influenza, and of these, two were of the pediatric population, aged between 0–1 years [7,9,10,12].

The total number of cases of acute respiratory infections in the first week of October 2022 (flu, pneumonia, and acute upper respiratory tract infections), increased compared to the previous year. We saw an unexpected increase in influenza infections following the two years (2020 to 2022) during which the virus had stopped spreading in our nation [13].

Oseltamivir usage is effective and secure in large observational studies [10,11,12,13,14]. Hospitalized adults showed substantial decreases in severe outcomes; however, these benefits were reduced and not statistically significant in children [13,14]. For many reasons including safety concerns, oseltamivir is still disputable in some groups [14,15,16].

Influenza is a difficult-to-control seasonal respiratory virus due to its high degree of infectivity and the severe potential it has in certain population groups (infants, children and the elderly with comorbidities, immunosuppressed) [17,18,19]. Trivalent or quadrivalent inactivated vaccines are the only effective means of preventing seasonal influenza infection [20,21]. When it comes to preventing influenza infection, chemoprophylaxis medication should not be utilized in place of the seasonal influenza vaccination [22,23,24]. The Romanian Ministry of Health advises using the influenza vaccine, especially for elderly individuals, small children, and those with weakened immune systems who are most at risk of influenza-related complications. The classic influenza-like sickness symptoms, including fever, sore throat, cough, headache, muscle and joint pain (especially in children), and severe malaise, are present in 20% to 40% of cases of influenza [25]. There are notable distinctions in flu symptoms in younger and older children, even though they may share many of the same symptoms. Infants and newborns can develop unexplained high temperatures [26]. Young children may develop convulsions, or febrile seizures, when their body temperature rises above 39.5 °C. In young children, the flu is a major contributor to lung infections, bronchiolitis, and croup. In older children, stomach upsets, vomiting, diarrhea, and abdominal discomfort are more prevalent, and earaches are also more frequently reported [27]. Moreover, severe clinical symptoms and complications, including rhinosinusitis, pneumonia, myocarditis, encephalitis, gastroenteritis, otitis, and acute respiratory distress syndrome, are typically associated with seasonal influenza infections in children. These outcomes lead to notably elevated rates of hospitalization and mortality in children [25,26,27,28,29].

The variations of specific influenza types in the pediatric population show a direct relationship with the symptoms in children. Higher attention was seen for influenza A symptoms in comparison with B influenza, due to the facts that the B virus presents a limited higher level of severity and could be less virulent than influenza type A. While influenza A and B viruses can lead to hospitalization and various difficulties in children, type A influenza is generally believed to cause more severe infections than type B viruses [30]. Children with influenza A are often younger than those with influenza B, and since the illness’ clinical presentation differs depending on age group, all results when comparing the severity of influenza A and B infections in children should be adjusted by age [28,29,30,31].

Oseltamivir, an oral administration neuraminidase inhibitor, has been the most frequently prescribed medication for the treatment of influenza in children since the year 2000 when it was approved by the Food and Drug Administration (FDA) [2,9,11,26,27,28,30,31]. As a prodrug of oseltamivir carboxylate, its mechanism of action is to inhibit the release of the virus from the infected host cells and to reduce the spread of influenza virus in the respiratory tract.

Neuraminidase inhibitors, such as zanamivir, laninamivir, and peramivir, have a rapid onset of action that leads to significant reductions in the duration and severity of symptoms. Treatment with a neuraminidase inhibitor has been demonstrated to be successful in reducing the risk of fatalities and serious sequelae if initiated within two days from the beginning of symptoms, and in certain cases, even later [4,6,7,8,9]. These inhibitors can be given within a one to two-day therapeutic window; effectively shortens the time taken for symptoms to subside [31,32,33,34]; and also lowers the rate of hospitalization and complications [30,31,32,33,34], such as pneumonia, acute respiratory distress syndrome, myocarditis, encephalitis, gastroenteritis, and acute otitis media. Antiviral-resistant variants of the virus have also been found in children receiving oseltamivir treatment [6,7,8,9,34,35,36].

Nevertheless, the influenza burden persists as a large number of youngsters are still unvaccinated, despite many vaccine recommendations from the World Health Organization for the risk groups in the population, including children aged 6 months to 5 years and people with specific comorbidities. In Romania, the Ministry of Health offers free of charge influenza vaccines for children aged 6 months to 59 months for eligible vaccination in family doctors’ offices and pharmacies. During the 2022–2023 and 2023–2024 influenza seasons, the tetravalent influenza vaccine was available, with the antigenic composition recommended by the WHO, but the coverage across all populations was very low [31,32,33,37]. Influenza diagnostic tests are crucial because they enable the confirmation of the illness’ existence and facilitate the implementation of targeted treatment [31,33,35,36,37,38].

Our study aimed to assess the clinical efficacy and tolerability of oseltamivir in children who had not received vaccinations and had various influenza virus types and subtypes in two consecutive influenza seasons in Romania, and to offer more information about the comorbidities associated with influenza in children.

## 2. Materials and Methods

We performed a single-center observational study of epidemiological, clinical, and adverse events recorded for children aged ≤18 years who were infected with influenza A or B and required hospitalization between 1 October 2022 and 30 May 2023 and 1 October 2023 and 4 May 2024. A clinical diagnosis of influenza was made based on the presence of a fever together with one or more of the following symptoms: sore throat, nasal congestion, pharyngeal erythema, upper respiratory tract symptoms (cough, rhinorrhea) headaches, myalgia, arthralgia, weakness, watery eyes, nausea, vomiting, and diarrhea. The clinical diagnoses of influenza in pediatric patients were confirmed with polymerase chain reaction (PCR) for influenza and included in the study. We utilized the AllplexTM Respiratory Panel 1 kit from Seegene in Seoul, Republic of Korea, and the GenXpert equipment from Cepheid, Sunnyvale, CA, USA, using real-time RT-PCR to detect influenza A and influenza B in the pediatric group included in this investigation. The Ethics Committee of the National Institute of Infectious Disease “Prof. Dr. Matei Bals”, located in Bucharest, Romania, approved all ethical problems, under registration number C03608/05/04/2024.

We extracted data for demographic characteristics such as age and gender, and information from past medical history, including co-morbidities, medications, symptoms, and signs of disease. We identified complications such as pneumonia, bronchitis, otitis, sinusitis, or other respiratory tract infections. When pediatric patients with comorbidities were hospitalized with influenza, we found that the outcomes varied. The following comorbidities were examined, concerning the severity of influenza illness requiring hospitalization and length of hospital stay. The main comorbidities were being overweight and class 1 obesity, dermatological diseases (atopic dermatitis and allergies), blood disease (anemia and thrombocytopenia), asthma and chronic respiratory illness, and renal and gastrointestinal disease.

When assessing the body fat in children aged 2 to 20 years old, body mass index (BMI), is calculated using the formula weight/height^2^; kg/m^2^) [37]. Oseltamivir was administered orally for 5 days, 2 × 75 mg/day for patients weighing ≥ 37.5 kg, and 2 mg/kg/day for those weighing < 37.5 kg.

The time for the onset of fever, the administration of oseltamivir, and the resolution of fever was recorded. A temperature of 37.5 °C was considered a fever. The patient’s body temperature was measured twice per day (8 a.m. and 8 p.m.). When we found a lower temperature than 37.5 °C and was maintained for 24 h, this was defined as the time when the patient became afebrile. The cure was defined as the remission of all signs and symptoms used for clinical diagnosis.

The following statistical measures were computed: means, medians, and ranges for continuous variables, and counts and percentages for categorical variables. The Chi-squared test was used to compare the frequency/percentage presentations of categorical variables, and the Student’s *t*-test, for continuous variables (n%). We performed a linear regression analysis to examine whether there was an association between the above variables and the duration of hospitalization in children with comorbidities. Multivariate logistic regression models included variables that were substantially correlated with the prescription of oseltamivir. Regardless of their correlation with the problems or prescriptions for oseltamivir in the univariate analysis, diabetes mellitus, obesity, chronic respiratory illness, high-risk patients, and age were included in the multivariate analyses. A *p*-value less than 0.05 was defined to indicate statistical significance. The statistical evaluation was performed in GraphPad Prism, version 9.5.1, USA.

Written informed consent was obtained from the parents of the minor children included in this study.

## 3. Results

A total of 1300 pediatric patients, who were enrolled in this study during the two following influenza seasons 2022–2023 and 2023–2024, met the inclusion criteria, tested positive for influenza and were all treated with oseltamivir. The demographic characteristics of the pediatric patients are summarized in Table 1. Of the group hospitalized in the 2022–2023 season, 89% (704/791) had influenza A viral infection with 86.4% of subtype A(H1N1)pdm09 and 13.6% of A(H3N2), and influenza B, with 11% (87/791) of the pediatric patients. Of the total patients, 59% were male and the median age was 2.4 years, IQR (1.02–9.28). In patients hospitalized in the 2023–2024 influenza season, 509 pediatric patients tested positive for influenza. Of these, 56.9% were of the male gender. The median age was 2.7 years, IQR (2.01–10.4). Of the total patients, 81.6% had influenza A, with 83.4% having subtype A(H1N1)pdm09 and 16.6% of patients having subtype A(H3N2). A total of 18.4% had influenza B. The patients arrived at the hospital at a median of two days (IQR, 1–4) following the beginning of symptoms. In the majority of the age categories, the ratio of males was significantly greater than that for female cases. The demographic data of the index cases in the pediatric population are listed in Table 1. Regarding the distribution in the pediatric population of the number of cases of influenza, the cases with influenza A(H1N1) were predominant (83.4%), and influenza B was more frequent in the 2023–2024 season for the 15–18 age group. There were differences in terms of the gender criteria, with males representing a percentage of 56.9% of total pediatric patients.

We found complications in the total group (e.g., gastrointestinal disease in 49.92%, pneumonia in 49.46%, bronchitis in 39.08%). These were differently distributed between patients, and during the 2023–2024 influenza season, the complications were lower than in previous group (see Table 1).

Looking at the duration of hospitalization among children with influenza, we found three comorbidities with significant association. Children with obesity, asthma, and gastrointestinal disease had median days of hospitalizations longer than children without comorbidities. The median days of hospitalization were statistically significant and longer among children with asthma (9 days vs. 4 days, *p* = 0.02); children with obesity were also significantly longer (7 days vs. 4 days, *p* = 0.001). The median hospitalization for children with influenza and gastrointestinal comorbidity was longer (5 days vs. 4 days, *p* = 0.0089). (see Table 2). 

We performed a linear regression to identify the association between influenza hospitalization and demographic characteristics (gender) of pediatric patients who had comorbidities. Children who were 5–14-years-old and 15–18-years-old were more likely to be admitted with influenza if they suffered from obesity (OR: 2.14; 95% CI: (0.21–20.42); *p* = 0.01 and OR: 12.85; 95% CI: (3,23–43.44); *p* = 0.03, respectively). We also found a statistical significance in children with asthma (OR: 12.34; 95% CI: (6.54–19.03); *p*-value = 0.0112), for the 5–14 age group. Other comorbidities that we analyzed were gastrointestinal disease for 2–4 years (OR: 12.75; 95% CI (5.33–13.21); *p* = 0.05) (see Table 3). Looking at gender, males who were 2–4-years-old and 5–14-years-old had a significant association with influenza hospitalization.

The comorbidities studied in this group were as follows:Metabolic and endocrine disease, such as diabetes and thyroid conditions.Respiratory such as viral pneumonia, bronchitis, and previous pulmonary disease.Gastrointestinal disease such as gastroesophageal reflux disease, irritable bowel syndrome, and dyspepsia.Renal disease including reno-ureteral malformations, and recurrent urinary tract infections. (See Table 3).

The treatment performed in the pediatric population with oseltamivir alone, and the one associated with the antibiotic and dexamethasone, had statistical significance when we compared this therapeutic combination between the types and subtypes from the two influenza seasons (2022–2023 and 2023–2024 groups). Patients who had changes in the laboratory analyses, including for C-reactive protein (CRP), showed significant differences in values for influenza A (i.e., AH1N1 season 2022–2023 vs. AH1N1 in 2023–2024 (*p* = 0.021)). Hemoglobin (*p* = 0.001), aspartate aminotransferase (AST) (*p* = 0.05), and alanine aminotransferase (ALT) showed significant differences for influenza B when compared with influenza A (*p* = 0.01).

The length of hospitalization was statistically significant in terms of the two consecutive seasons for both influenza A and B. We evaluated the median duration of hospitalization and recovery rates among the two groups treated with oseltamivir (see Table 4).

In the 2022–2023 group, 6.19% of the 509 hospitalized children with obesity were hospitalized for a median of 14 days, vs. 7 days for 16.9% of the 2023–2024 group of 509 pediatric patients (*p* = 0.001).

The median duration of hospitalization was also shorter in the oseltamivir group from the 2023–2024 influenza season with asthma than in the similar 2022–2023 group (*p* = 0.05).

After the onset of oseltamivir, all pediatric patients recovered favorably and no child died. After initiating treatment with oseltamivir, it was not necessary to interrupt its administration due to adverse reactions. Even if these appeared during the treatment (diarrhea 9% and nausea 11% of the total group), they were tolerated after the administration of the symptomatic medication.

## 4. Discussion

In our observational study, we enrolled a large number of pediatric patients, divided into influenza A and influenza B, from two consecutive seasons after the COVID-19 pandemic. This study showed that a significant proportion of the hospital admissions in pediatric patients are caused by influenza. The efficacy of oseltamivir was compared and showed a reduction in the duration of illness and hospitalization. Regardless, the advantages of using oseltamivir need to exceed the disadvantages in terms of side effects and efficacy, as our study demonstrates. As the pandemic years and the COVID-19 illness have passed, it seems that the number of complications in pediatric influenza cases has decreased. However, the number of complications is higher than in the previously published studies, possibly as a result of having noted that all of the included patients in both groups who were studied in our clinic were not vaccinated against influenza [31,32,33,34,37,38,39,40]. Our pediatric patients spent a median of four days in the hospital, for patients without comorbidities, and all of them were administered antiviral therapy while hospitalized. This study examined the correlation between the severity of influenza-related hospitalization and the duration of hospital stay among the following comorbidities: endocrine or metabolic disease; asthma; respiratory, renal, gastrointestinal, and blood disease; and eczema/atopic dermatitis. We also found that the longest hospitalization was in pediatric patients with asthma and influenza. Similarly, other studies reported an increased length of stay in hospital due to comorbidities, including asthma [28,30,32]. When we compared the pediatric patients without comorbidities for the duration of hospitalization, there was no evidence that oseltamivir exacerbated participants’ asthmatic bronchoconstriction. On the other hand, data showed that hospitalized children from the 2023–2024 influenza season fully recovered faster in the oseltamivir-treated group than in the group from the 2022–2023 influenza season [16,19,26,28,39,41]. We found that compared to other age groups, children who were obese and in the 5–14 and 15–18-year-old age groups had higher admission rates for influenza. Similar studies revealed that obesity comes independently as a risk factor for COVID-19 and other viral infections, as well as for the severity of these illnesses [41,42,43]. We recorded the median duration of hospitalization and recovery rates among those treated with oseltamivir alone, and patients treated with oseltamivir associated with dexamethasone alone, or in combination with dexamethasone and antibiotics between two consecutive influenza seasons. In the group with influenza B, the recovery rates and duration of hospitalization were the longest, and with an increase in the 2022–2023 influenza season. The recovery rate was greater in the 2023–2024 group than in the 2022–2023 group for the oseltamivir group with comorbidities, such as asthma, chronic pulmonary disease, and obesity (*p* = <0.05), which was similar to other studies [28,31,41,42,43]. The following comorbidities were the most common in the study group: respiratory-related diseases, followed by nutritional and metabolic diseases (obesity). Our study provides the largest comparative analysis of oseltamivir using a large cohort of children as the study group, with laboratory-confirmed influenza. Data from our study, based on hospitalization registers, reported the incidence of influenza in children below 18 years of age. Additionally, it should be mentioned that the data from this study can help make national decisions to reduce the risk of influenza complications, especially in children with comorbidities. One of the most important ways to reduce the risk is to prevent influenza by vaccination, with special attention in young children. Previous studies have revealed cases of dual influenza and bacterial co-infections in children [17,20,26]. In pediatric populations, co-infections between influenza and bacterial illnesses have been shown to happen 30–50% of the time [31,34]. A subsequent bacterial infection frequently coexists with or follows an influenza illness [21,22]. In this situation, antiviral and antibiotic treatments are advised to treat the co-infection of influenza and bacteria [32]. Because of this, the subgroup analyses of hospitalized children who received anti-influenza medication or who did not obtain conventional antibiotics were the main focus of our subsequent subgroup studies. There were no new safety concerns found, and the safety profile shown in children under the age of 18 was very comparable to those previously seen in other studies [6,12,16]. Our findings support earlier research [5,15,17,20] in showing that younger children had a higher risk of respiratory virus infection than older children. The influenza virus was discovered in children between 2–4 years of age more frequently than in any other age group. As mentioned earlier, the majority of children infected with influenza A or B, had symptoms including fever, myalgia, rhinitis, and cough [16,17,18,19], while the reports of concurrent gastrointestinal symptoms were apparent in influenza B patients for the age category 5–14 and 15–18 years old [20]. Furthermore, in line with other studies, we did find apparent variations in the clinical symptoms of the various viruses. Clinical similarities between children infected with influenza A or B were found in similar studies [30]. Effective preventative measures and treatment options are available against influenza viruses. Every year, a potent influenza vaccination becomes available. Nonetheless, only 7% of the Romanian population becomes vaccinated, both in adult and pediatric populations [38]. Oseltamivir, an effective antiviral medication, can be provided to children with influenza virus infection if it is discovered during the first 48 h of symptoms or if hospitalization is necessary [26]. Through follow-up studies, the clinical safety of oseltamivir has been thoroughly examined. Oseltamivir is suitable for treating influenza in pediatric populations; no significant safety concerns have emerged that limit its use. Similarly, when given for the purpose of chemoprophylaxis against influenza in adolescents, oseltamivir is well tolerated [31,33,41,42]. Oseltamivir is a valuable supplement to current influenza treatments for the control of influenza in communities, as all of the influenza virus isolates had normal inhibition of neuraminidase activity to oseltamivir. Additionally, the effectiveness of the treatment in the pediatric population, from one season to another, is obvious. In a systematic review and meta-analysis previously published, the results obtained after the administration of oseltamivir reduced the duration of hospitalization and was associated with a significant reduction in mortality [43]. We found studies that have described the clinical outcomes in children with influenza when compared to a placebo or conventional treatment, and the results showed that oseltamivir may shorten hospital stays in individuals with severe influenza, which is similar to our results [41,42,43,44]. Due to the low vaccination rate among pediatric patients in our nation, it was not possible to offer a comparative cohort of vaccinated children in our study, which presents some disadvantages. This comparative study did not include other antiviral medicines, however, this research will continue in the upcoming influenza virus season. Another research limitation was the absence of a drug resistance test for influenza antiviral agents using genotypic and phenotypic analysis, considering that the duration of hospitalization in patients with fever after oseltamivir administration was longer for patients with influenza A(H1N1) infection than influenza A(H3N2), in both seasons [36,41,42,43,44,45].

The outcomes of oseltamivir treatment in hospitalized patients with detectable influenza virus infections at the time of admission can be shown by our results and those of other comparable studies [40,41,42,43,44,45]. The relative contribution of influenza treatment in the presence of comorbidities can be estimated using an approach that enrolls patients after their illness has become severe enough to require hospitalization. Future complex studies will bring even more information to the current knowledge.

## 5. Conclusions

Our findings suggest that influenza predominantly impacts children, that virus strains in each season may be slightly different, and that the influenza hospitalization incidence among children aged <18 years in Romania can be related to age, comorbidities, seasons, duration of hospitalization, and symptom differences. The increasing trend in hospitalizations in recent years, which was observed in this study, indicates the urgent need to focus on influenza vaccination, especially in children with comorbidities. A long-term epidemiological study is needed to understand the changing features of influenza in Romania.

## Figures and Tables

**Table 1 viruses-16-01576-t001:** Characteristics of patients with RT-PCR results confirmed for influenza; categorical data: n (%).

Characteristics	Overall n (%) 1300 (100)	Influenza A(H1N1) 2022–2023 Season n (%) 608 (86.4)	Influenza A(H3N2) 2022–2023 n (%) 96(13.6)	Influenza B 2022–2023 n (%) 87(11)	Influenza A(H1N1) 2023–2024 n (%) 346 (83.4%)	Influenza A(H3N2) 2023–2024 n (%) 69(16.6%)	Influenza B 2023–2024 n (%) 94(18.4%)	*p*-Value
Age: years old								0.12
0–1	161 (12.4)	60 (9.8)	5 (5.6)	2 (2.7)	39 (11.3)	6 (8.9)	2 (1.7)	
2–4	537 (41.3)	227 (37.3)	27 (28.3)	13 (14.6)	92 (26.5)	16 (23.6)	12 (13.2)	
5–14	338 (26)	141 (23.2)	34 (35.2)	34 (38.8)	120 (34.7)	23 (32.8)	42 (44.2)	
15–18	264 (20,3)	180 (29.7)	30 (30.9)	38 (43.9)	95 (27.5)	24 (34.7)	38 (40.9)	
Gender (%)								0.12
Female	560 (43.1)	281 (46.2)	44 (45.3)	37 (42.9)	150 (43.5)	31 (44.3)	40 (42.1)	
Male	740 (56.9)	327 (53.8)	52 (54.7)	50 (57.1)	196 (56.5)	38 (55.7)	54 (57.9)	
Symptoms								
Fever	1222 (94%)	600 (98.68) *	85 (88.54)	80 (91.95) #	314 (90.75)	63 (91.3)	80 (85.1) #	0.05 #; 0.01 *
Cough	1157 (89%)	598 (98.36) *	85 (88.54)	47 (54.02)	312 (90.17)	60 (87)	55 (58.51)	0.13; 0.01 *
Weakness	832 (64%)	342 (56.25)	49 (51.04)	74 (85.05)	262 (75.72)	27 (39.13)	78 (83)	0.10
Nasal congestion	598 (46%)	218 (35.86)	53 (55.2)	51 (58.62)	181 (52.31)	30 (43.48)	65 (69.15)	0.14
Sore throat	728 (56%)	345 (56.74) *	44 (45.83)	43 (49.43)	182 (52.6)	31 (45)	64 (68)	0.2; 0.005 *
Headache	562 (43.23%)	189 (14.54)	40 (41.67)	32 (36.78)	203 (58.67)	42 (60.86)	56 (59.57)	0.3
Myalgia	561 (43.15%)	132 (21.71) *	35 (36.46)	80 (91.95)	231 (66.76)	43 (62.32)	40 (42.55)	0.07; 0.0011 *
Watery eyes	375 (28.85%)	182 (29.93)	23 (24)	15 (17.24)	102 (29.5)	21 (30.43)	32 (34.04)	0.13
Nausea	272 (20.92%)	107 (17.6)	12 (12.5)	21 (24.14)	80 (23.12)	27 (70)	25 (27)	0.12
Vomiting	677 (52.08%)	308 (50.66)	32 (33.33)	42 (48.28) #	210 (60.7)	29 (42.03)	56 (59.57) #	0.01 #
Diarrhea	575 (44.23%)	274 (45.07)	21 (21.88)	74 (85) #	109 (31.5)	32 (46.38)	65 (69.15) #	0.05 #
Complications n (%)								
Pneumonia	643 (49.46)	328 (53.95)	56 (58.33)	76 (87.36)	87 (25.14)	42 (60.86)	54 (57.45)	
Sinusitis	440 (33.84)	187 (30.76)	46 (47.92)	44 (50.57)	121 (34.97)	14 (20.29)	28 (29.79)	
Bronchitis	508 (39.08)	279 (45.89)	35 (36.46)	39 (44.83)	101 (29.19)	22 (31.88)	32 (34.04)	
Pharyngitis	407 (31.31)	126 (20.72)	67 (69.79)	70 (80.46)	110 (31.79)	25 (36.23)	9 (9.57)	
Acute otitis media	417 (32.08)	200 (32.89)	30 (31.25)	39 (44.83)	98 (28.32)	26 (30.43)	24 (25.53)	
Gastrointestinal disease	649 (49.92)	309 (50.82)	55 (57.29)	48 (55.17)	180 (52.02)	21 (30.43)	36 (38.29)	

Fever (94%) and cough (89%) were the most commonly reported symptoms, and myalgia was associated with influenza B in the 2022–2023 season. The presence of fever, vomiting, and diarrhea was associated with statistical significance in the multivariate analysis (*p* < 0.001) in patients with influenza B, in both seasons, for clinical diagnosis (#). Influenza A (H1N1) diagnosis was associated with fever, cough, sore throat, and myalgia (*) (*p* < 0.05).

**Table 2 viruses-16-01576-t002:** The association of different comorbidities with influenza-related duration of hospitalization.

Median Days of			
	Hospitalizations (IQR)	Median Days (IQR)	*p*-Value
Comorbidities	No	Yes	
Obesity	4 (1–5)	7 (5–7)	0.001
Endocrine or			
metabolic disease	3 (1–4)	4 (2–6)	0.79
Asthma	4 (1–4)	9 (8–10)	0.02
Respiratory disease	3 (2–4)	4 (2–6)	0.077
Blood disease	2 (1–6)	3 (2–5)	0.78
Gastrointestinal			
disease	4 (1–6)	5 (1–5.5)	0.0089
Eczema/			
atopic dermatitis	2 (1–3)	3 (1–7)	0.46
Renal disease	4 (1–6)	6 (2–7)	0.35

Note: Data are presented as the median (interquartile range). We did not perform group comparisons for variables with overall *p*-values > 0.05. *p*-values ≤ 0.05 were considered statistically significant. Abbreviations: IQR, interquartile range.

**Table 3 viruses-16-01576-t003:** The association between the comorbidities and hospitalizations for influenza in children, by age group and gender. OR (95% CI), *p*-value.

Variables	0–1 Years Old	2–4 Years Old	5–14 Years Old	15–18 Years Old
Comorbidities				
Obesity	-	2.21 (4.23–13.32);0.5	2.14 (0.21–20.42); 0.01	12.85 (3,23–43.44); 0.03
Endocrine/metabolic disease (1)	-	-	3.14 (2.456–9.012); 0.21	-
Asthma	-	-	12.34 (6.54–19.03); 0.0112	9.67 (3.23–22.75); 0.07
Respiratory disease (2)	-	14.44 (14.92–23.12); 0.11	2.52 (0.15–13.24); 0.10	12.24 (8,68–21.75); 0.123
Gastrointestinal disease (3)	9.31 (4.244–15.65); 0.123	12.75 (5.33–13.21); 0.05	14.43 (12.23–32,33); 0.5	2.33 (6.01–12.07); 0.15
Renal disease (4)	-	-	12 (2.90–22-07); 0.77	-
Atopic dermatitis	1.23 (1.23–5.96);0.09	12.44 (8.97–19.23); 0.86	-	1.23 (2.12–21.05); 0.6
Gender Male Female	1.44 (2.01–11.03); 0.14 10.95 (3.47–34.3); 0.12	15.45 (2.59–89.87); 0.03 11.24 (8.47–19.97); 0.5	8.63 (2.34–3.01); 0.01 6.91 (1.44–13.23); 0.12	15.56 (3.34–23.98); 0.9 8.77 (2.25–31.32); 0.5

Note: Standard statistical significance is applied to values that are bolded. *p*-values ≤ 0.05 were considered statistically significant.

**Table 4 viruses-16-01576-t004:** Characteristics of the patients, observed by RT-PCR, treatment, and laboratory parameters; categorical data: n (%).

Parameters	Influenza A (H1N1) 2022–2023 Season n (%) 608 (86.4)	Influenza A (H3N2) 2022–2023 n (%) 96 (13.6)	Influenza B 2022–2023 n (%) 87 (11)	Influenza A (H1N1) 2023–2024 n (%) 346 (83.4%)	Influenza A (H3N2) 2023–2024 n (%) 69 (16.6%)	Influenza B 2023–2024 n (%) 94 (18.4%)	*p*-Value
Treatment n (%)							
Duration of hospitalization days n (IQR)	5 (3–6)	4 (2–5)	7 (1–7)	5 (4–6)	4 (3–5)	6 (2–6)	0.01
Oseltamivir	402 (66.12)	59 (61.46)	39 (44.83)	214 (61.85)	17 (24.64)	18 (19.15)	0.001
Oseltamivir and dexamethasone	124 (20.4)	22 (22.91)	31 (35.63)	49 (14.16)	29 (42.65)	14 (14.89)	0.11
Oseltamivir, dexamethasone and antibiotics	82 (13.49)	15 (15.63)	27 (31.03)	83 (24)	41 (42.03)	32 (34.04)	0.001
Laboratory n (%)							
AST, U/L > 5× normal values	124 (20.39)	76 (60.32)	38 (67.86)	24 (6.94)	12 (17.39)	36 (53)	0.05
ALT, U/L > 5× normal values	136 (22.37)	89 (92.71)	45 (51.72)	42 (12.14)	43 (62.32)	21 (22.34)	0.01
Hemoglobin	216 (35.53)	70 (55.5)	36 (64.3)	22 (6.36)	29 (42.03)	32 (47.1)	0.001
Leukocyte	112 (18.42)	67 (69.79)	65 (74.71)	24 (6.94)	56 (81.16)	18 (19.15)	0.31
Lymphopenia	98 (16.12)	34 (27)	21 (37.5)	14 (4.05)	17 (24.64)	12 (17.6)	0.32
Neutrophilia	109 (17.93)	23 (23.96)	11 (12.64)	32 (9.25)	23 (33.33)	13 (13.83)	0.1
Low Thrombocyte	98 (16.12)	37 (29.4)	14 (25)	33 (9.54)	35 (50.74)	21 (30.9)	0.11
High Prothrombin time	123 (20.23)	76 (60.32)	44 (78.6)	21 (6.07)	29 (42.03)	30 (44.12)	0.23
LDH elevated	234 (38.49)	51 (40.5)	28 (50)	40 (11.56)	32 (46.38)	22 (32.35)	0.31
CRP elevated	112 (18.42)	55 (43.65)	32 (57.14)	42 (12.14)	44 (63.77)	21 (30.88)	0.021

## Data Availability

The datasets generated and analyzed during the current study are available from the corresponding author upon request.

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
