# Peer review of "The Clinical Effectiveness and Tolerability of Oseltamivir in Unvaccinated Pediatric Influenza Patients during Two Influenza Seasons after the COVID-19 Pandemic: The Impact of Comorbidities on Hospitalization for Influenza in Children"

_viruses, 2024, doi:10.3390/v16101576_

Round 1

Reviewer 1 Report (Previous Reviewer 1)

Comments and Suggestions for Authors

Thank you for your trust and the opportunity to review the manuscript again titled now "  

 The Clinical Effectiveness and Tolerability of oseltamivir in Unvaccinated Influenza Pediatric Patients in Two Influenza seasons after the COVID-19 pandemic. The Impact of Comorbidities on Hospitalization for Influenza in Children. "

The article has been thoroughly revised.

The authors have removed the critical Table 4 ( regarding flu symptoms among children) and have added flu symptoms without division into the age of the examined children to Table 1.

The authors have clarified doubts.

The authors have added new paragraphs.

Author Response

Dear Reviewer 1, thank you very much for taking the time to review this manuscript.

The article was improved with your help, we tried to correct it as you instructed us in the previous version. thank you for your effort and advice. Best regards.

Reviewer 2 Report (New Reviewer)

Comments and Suggestions for Authors

1. The footnotes for Table 2 have not been included. 

2. The title of Table 3 is bellow the table and it must be above Table 3.

Author Response

Dear Reviewer 2,

Thank you very much for taking the time to review this manuscript. Please find the responses below and the corrections highlighted in the re-submitted files.

For table 2 we added the notes between line 231-234.

We modified the Titles for both table 3 and 4 and added above it. We appreciate the recommendations, and we hope to have succeeded in making the requested changes.

Thank you!

Reviewer 3 Report (Previous Reviewer 3)

Comments and Suggestions for Authors

I have reviewed the revised manuscript and decided to reject this paper because of no changes of main points.

Author Response

Thank you very much for taking the time to review this manuscript. 

We hope to find the improvements in the last form of our manuscript. 

This manuscript is a resubmission of an earlier submission. The following is a list of the peer review reports and author responses from that submission.

Round 1

Reviewer 1 Report

Comments and Suggestions for Authors

Thank you for the opportunity to review the manuscript titled, "  The Clinical Effectiveness and Tolerability of Oseltamivir in Unvaccinated against Influenza children; a Romanian comparative study between 2022-2023 and 2023-2024 Influenza seasons. "

The framework of this article was planned correctly.

There are some small issues that the authors should address before the manuscript can be considered for publication.

The following are my comments describing these issues.

1. Introduction

Line 80- The classic influenza-like sickness symptoms….Do these flu symptoms affect children or adults?

We know that the flu affects kids more severely than adults. In children, the fever associated with influenza is likely to be higher, on average, than an adult's fever. In addition, children more often experience symptoms related to the digestive system, such as nausea, vomiting and diarrhea, but the authors did not mention about it  (but the authors mentioned about gastrointestinal symptoms in Table 4).

The authors should have presented the symptoms of influenza among children, divided into younger and older children. Please correct this.

The authors should also describe the symptoms of influenza A and B and pay attention to which symptoms dominated in the general population in the 2022/2023 season and which in the 2023/2024 season.

Type A influenza is generally considered worse than type B influenza. This is because the symptoms are often more severe in type A influenza than in type B influenza.

Please complete this.

Line 111- Nevertheless, the influenza burden persists as a large number of youngsters are still unvaccinated despite many vaccine recommendations. Where ? When?

Please add and complete the references

What was the aim of the study? Please add it.

2. Result

Table 4.

Influenza A(H1N1)group 2022-2023/group 2023/2024 n

what does “n” mean?

Table 4 How do the authors know that children in aged 0-1 have symptoms of myalgia or/and malaise? Please explain this.

Author Response

Thank you for your time and consideration, we followed the instructions and hope to send a better version of the manuscript. 

Best regards, 

Reviewer 2 Report

Comments and Suggestions for Authors

The article presents the results of a study assessing the effectiveness of oseltamivir among children aged ≤18 years infected with influenza A or B viruses that required hospitalization between 1st October 2022 and 30th May 2023 and 1st October 2023 and 4th of May 2024. The article is difficult to follow because the methodology is not explained in sufficient detail and the text includes comments explaining the results obtained that could not be find in the tables. The methodology of the statistical analysis is not explained in sufficient detail, and the article presents unclear statistical analysis. 

It is necessary to solve the problems detected. The research and the article have the following problems:

1. Methods section. The Methods section does not explain what comorbidities were assessed in the study and how they were defined and detected. This is a critical weakness of the Methods section because a number of comorbidities are included in Table 3.

2. Lines 138-142. These two paragraphs does not explain in a clear way the statistical analysis carried out in the study. It does not explain how normal an non-normal variables were detected, which of the presented tests were used in each situation, and the p values considered. Table 3 presents ORs, but the Methods section does not explain this.

3. Line 161. “The gender criteria do not bring significant differences.” This comment is incorrect from a statistical point of view. For example, the overall percentage of male cases was significantly greater than that for female cases, with P< 0.0001 and OR= 1.74. The same comment for other comparisons between males and females.

4. Lines 171-175. The comment presented in this paragraph is incorrect. Table 3 does not present the comparison for different laboratory outcomes in cases treated with Oseltamivir and with Antibiotic and Dexamethasone. Consequently, it is incorrect to comment that these two groups had statistical significance for patients who had changes in the laboratory analyses including C-reactive protein (CRP) (p=0.021), Hemoglobin (p=0.001), Aspartate Amino Transferase (AST)(p=0.05), and Alanine Amino Transferase (ALT) (p=0.01).

5. Lines 176-177. “We evaluated the median duration of hospitalization and recovery rates among those two groups treated with oseltamivir.“ What are the two groups mentioned in this sentence?

6. Lines 177-179. “In the 2022-2023 group, 20.8% of 509 hospitalized children with comorbidities were hospitalized with a median of 14 days, vs 7 days for 2023-2024 group of 791 pediatric patients. (p=0.001).” This comment is confusing and it seems incorrect. The first row is about cases with obesity in the two periods, and they are 136 cases in both periods instead of 1300. The comparison of medians can be correct, but the OR indicates that the differences were not significant. The Methods section does not explain why ORs were used to compare the median hospitalization duration, and it is impossible to know what is the meaning for the ORs presented in this table.

7. Table 3. There are doubts about the analysis presented in this table for several reasons. The Methods section does not explain how the analysis presented in this table was carried out. The category “Any comorbidity” includes 225 cases, but the comorbidities considered in the study are not explained in the Methods section and in table 3. The comparison of medians was statistically significant for chronic pulmonary patients, but the hospitalization duration was the same in both groups. Consequently, both medians cannot be different.

8. Lines 185-187.  “Comorbidities were present in 10.38% of obesity influenza‐confirmed admissions and were associated with more severe outcomes in group season 2022-2023 and longer hospitalization.” This comment is strange because obesity seems to be considered a comorbidity, based on Table 3. Additional explanations are necessary to clarify this comment. For example, what are the more severe outcomes present in cases with obesity that are not present in non-obese cases?   

9. Lines 187-189. “Children with comorbidities were more likely to experience severe influenza with ICU admission for chronic pulmonary disease and the longest hospitalization, median of 13 days, with IQR (6-16 days)”. This comment is incorrect. The median hospitalization of 13 (6-16) corresponds to cases with chronic pulmonary disease in 2023-2024, not to children with comorbidities.

10. Table 4. It is unclear what was assessed in Table 4. The Methods section does not explain how the prevalences of symptoms in different age groups and seasons were compared. It is unclear what does it mean a significant p value in table 4 because this table does not present the prevalence of symptoms in different age groups and seasons.

Comments on the Quality of English Language

The quality of English must be improved.  

Author Response

Thank you for your time and recommendations. we hope the manuscript is in a better form, as you requested.

Best regards, 

Reviewer 3 Report

Comments and Suggestions for Authors

In this paper, the authors conducted a retrospective study regarding clinical and laboratory characteristics of childhood seasonal influenza occurred in the 2022/23 and 2023/24 year after COVID pandemic. They found that there were some variations of clinical severity according to influenza viruses and epidemic years and described that oseltamivir was effective on influenza virus infections.

The manuscript is relatively well-written with some data. However, this version has limitations to be published, because there are no new information, improper study design, and difficulty of interpretation of the data.

- Although the authors titled and described “the clinical effectiveness and tolerability of oseltamivir…” throughout this manuscript, this study was not a comparative study for effectiveness of antiviral and all subjects were received oseltamivir. Additionally, the subjects are included children with various underlying diseases such as asthma, chronic pulmonary diseases, and other diseases that can have long morbidity. Thus, there are confounding factors to estimate hospitalization and severity of the disease since the number of hardscaped patients influence on the results, and other pathogen (viruses) infections could affect these patients during influenza season (PCR positive is not a confirmative diagnostic, and there are PCR positive carriers with other respiratory pathogen infection).

- It is well known that oseltamivir has a limited effect on influenza-associated pneumonia and acute respiratory distress syndrome (ARDS) (PMID: 24811411). Also, the authors should know the issues that the pathophysiology of influenza and COVID-19 remains to be further evaluated, and that early immune modulators are needed in severe cases (PMID: 20822853, PMID: 32664709). Thus, they could change the study design with additional data analysis.

For example, Epidemiological changes (or observations) of seasonal influenza after COVID-19 pandemic.

Background: We evaluate epidemiological characteristics of 2 seasonal influenza in children (or all admitted patients. Method: During 2022/23 and 2023/24 (if possible, included COVID-19 pandemic years), yearly cases, monthly cases (winter season), age, sex, cases of severe symptoms (pneumonia/ARDS or hospitalization >10 days,), previously healthy cases and those with underlying disease, and others. Results: There are fluctuated cases in each year, and there were influenza A and influenza B cycle in an influenza season (by monthly case analysis, not simultaneously spread both viruses) and more severe cases (pneumonia) in influenzas B (or not); after age distribution of cases of healthy children, including asthma and obesity (by using figures not using table), influenza B is more prevalent in young children (or not). The proportion of cases of pneumonia (x-ray confirmed, and as a representative of severe cases) in both influenza season were similar (or not), and that of pneumonia in influenza B and A were similar (or not). Older children had more pneumonia than young children (or not), except infant (bronchiolitis cases), etc.  Conclusion. Our finding suggest that influenza affect mainly children, and viral strains in each season may be somewhat different as one of microbiota of human species. Long-term epidemiological study is needed for changing characteristics of influenza in Romania.

Author Response

Dear editors and reviewers, 

thank you for pointing out all the requests. We hope this time will be a better version, accordingly to publication remarks. 

Round 2

Reviewer 2 Report

Comments and Suggestions for Authors

The quality of the article has been improved. There are several aspects that require a clarification.

1. Lines 166-167 (tracked article). "We identified specific comorbidities associated with influenza in hospitalized children". This sentence must be reviewed because the study did not assess the association between comorbidities and influenza, but different outcomes in patients with influenza who had comorbidities.  

2. Lines 167-170. This paragraph must be reviewed. This paragraph must explain the comorbidities included in the study and, if that is possible, how were they detected. Obesity is not included in this paragraph, despite it was included in the study. It is necessary to include the definition used in the study for obesity. 

3. Lines 177-179. This explanation is incorrect. The t-test has not been used in to assess the results presented in Table 4. The t-test is used to compare means and the results in Table 4 are counts. The Methods section must explain what was assessed in table 4, the statistical test used and p values considered. 

4. Lines 240-242. "Children with comorbidities were more likely to experience severe influenza with ICU admission for chronic pulmonary disease and the longest hospitalization, median of 13 days, with IQR (6-16 days)." This sentence must be reviewed and clarified. Table 3 can show that children with comorbidities have longer hospitalizations than those without them, but you must include the hospitalization duration in patients without comorbidities for comparison.  Table 3 does not show that they are more likely to experience severe influenza with ICU admission for chronic pulmonary disease, because the ICU admissions were not mentioned in the article. The Methods section does not indicate that the ICU admissions were included in the analysis. 

Comments on the Quality of English Language

moderate editing of English is necessary

Author Response

we thank you for the recommendations,

We kindly ask for english correction, from you, please add it to our invoice. 

Best regards,

Carmen Pavelescu&team

Reviewer 3 Report

Comments and Suggestions for Authors

I have reviewed revised manuscript.

The authors insisted  main study design and result patterns although  there were some revisions in contents. I think that this version has limitations to be published. 

Author Response

Dear reviewer, thank you for your time and advice. Considering your answer, we appreciate that you sent us this feedback. We have added changes requested by the other reviewers; we hope the new version will be in the right direction from your perspective.  We had a very large batch, and it is clear that some limitations are in this study, but it will be the support for future research. Thank you for your understanding, 

Best regards.